# CD38 Deficiency Alleviates Diabetic Cardiomyopathy by Coordinately Inhibiting Pyroptosis and Apoptosis

**DOI:** 10.3390/ijms242116008

**Published:** 2023-11-06

**Authors:** Ling-Fang Wang, Qian Li, Ke Wen, Qi-Hang Zhao, Ya-Ting Zhang, Jia-Le Zhao, Qi Ding, Xiao-Hui Guan, Yun-Fei Xiao, Ke-Yu Deng, Hong-Bo Xin

**Affiliations:** National Engineering Research Center for Bioengineering Drugs and the Technologies, Institute of Translational Medicine, Nanchang University, Nanchang 330031, China; liqian_19961029@163.com (Q.L.); wen_k1995@163.com (K.W.); 15848149296@163.com (Q.-H.Z.); Zhangyating223@163.com (Y.-T.Z.); zjl199805@163.com (J.-L.Z.); dq1626377859@163.com (Q.D.); gxh_ncu@foxmail.com (X.-H.G.); xyf.84@163.com (Y.-F.X.); dky@nuc.edu.cn (K.-Y.D.)

**Keywords:** CD38, diabetic cardiomyopathy, pyroptosis, apoptosis, Sirt3

## Abstract

Diabetic cardiomyopathy is one of the diabetes mellitus-induced cardiovascular complications that can result in heart failure in severe cases, which is characterized by cardiomyocyte apoptosis, local inflammation, oxidative stress, and myocardial fibrosis. CD38, a main hydrolase of NAD^+^ in mammals, plays an important role in various cardiovascular diseases, according to our previous studies. However, the role of CD38 in diabetes-induced cardiomyopathy is still unknown. Here, we report that global deletion of the CD38 gene significantly prevented diabetic cardiomyopathy induced by high-fat diet plus streptozotocin (STZ) injection in CD38 knockout (CD38-KO) mice. We observed that CD38 expression was up-regulated, whereas the expression of Sirt3 was down-regulated in the hearts of diabetic mice. CD38 deficiency significantly promoted glucose metabolism and improved cardiac functions, exemplified by increased left ventricular ejection fraction and fractional shortening. In addition, we observed that CD38 deficiency markedly decreased diabetes or high glucose and palmitic acid (HG + PA)-induced pyroptosis and apoptosis in CD38 knockout hearts or cardiomyocytes, respectively. Furthermore, we found that the expression levels of Sirt3, mainly located in mitochondria, and its target gene FOXO3a were increased in CD38-deficient hearts and cardiomyocytes with CD38 knockdown under diabetic induction conditions. In conclusion, we demonstrated that CD38 deficiency protected mice from diabetes-induced diabetic cardiomyopathy by reducing pyroptosis and apoptosis via activating NAD^+^/Sirt3/FOXO3a signaling pathways.

## 1. Introduction

With changes of lifestyle and dietary habits, the prevalence of diabetes patients has increased rapidly all over the world. It is predicted that about 700 million people will suffer from diabetes by 2045 [1]. Diabetic cardiomyopathy is one of the severe complications of type 2 diabetes. It is characterized by adverse structural remodeling and abnormalities of cardiac function, which is not directly attributable to hypertension, valve disease, or coronary artery disease [2]. At present, specific strategies for preventing diabetic cardiomyopathy have not yet been established since the pathophysiological mechanisms of diabetic cardiomyopathy are far from being elucidated. Studies have indicated that cell death is the terminal pathway of cardiomyocytes in diabetic cardiomyopathy, in which different mechanisms may be involved in the process of cell death [3]. It has been reported that chronic hyperglycemia can induce myocardial apoptosis in patients with diabetes [4]. Tissue biopsies have also shown that cardiomyocyte apoptosis in diabetic patients was 85 times higher than in non-diabetic patients, suggesting that myocardial cells in diabetic patients were sensitive to cell death [5]. In addition to apoptosis, studies have also shown that pyroptosis participates in the process of diabetic cardiomyopathy [6,7]. Moreover, it has been reported that NOD-like receptor family pyrin domain containing 3 (NLRP3) inflammasome–caspase-1-mediated pyroptosis was presented in the myocardium of diabetic rats, while silencing of NLRP3 ameliorated cardiac inflammation and pyroptosis, improving myocardial function [8].

Nicotinamide dinucleotide (NAD^+^) as a key cofactor plays important roles in metabolism-driven disease progression [9]. It has been reported that NAD^+^ redox imbalance in the heart exacerbated diabetic cardiomyopathy [10] and nicotinamide phosphoribosyltransferase (NAMPT), the rate-limiting enzyme in the salvage pathway of NAD^+^ synthesis, alleviated the development of diabetic cardiomyopathy via decreasing oxidative stress [11]. In addition to NAD^+^, the gaseous signaling molecules nitric oxide and carbon monoxide were also critically involved in cardiovascular regulation. It was reported that carbon monoxide increased inducible NOS expression that mediated CO-induced myocardial damage during ischemia/reperfusion injury [12]. In addition, endothelial dysfunction is an important feature of diabetic cardiomyopathy and has been attributed to dysregulation of NO generation and bioavailabilty [13,14]. It has been reported that empagliflozin prevented cardiomyopathy via the soluble guanylate cyclase enzyme (sGC)-cGMP-PKG pathway in type 2 diabetes mice [15]. Therefore, the application of molecules targeting the NO-sGC-cGMP pathway in cardiovascular diseases, especially in diabetic cardiomyopathy, is also promising.

CD38 is the main hydrolase of cellular NAD^+^, and widely expressed in multiple tissues. Studies showed that the level of NAD^+^ was increased in the heart, brain, and other tissues of CD38 knockout mice [16]. In addition, the activity of NAD^+^-dependent deacetylase silent-mating type information regulation 2 homolog 1 (Sirt1) was significantly increased [17]. Our previous study also showed the increased level of NAD^+^ in CD38-deficient hearts protected the hearts from high-fat-diet-induced oxidative stress [18]. Numerous studies have shown that CD38 is involved in the pathological processes of many diseases including aging, obesity, cardiovascular diseases, cancer, and inflammation [19]. Our previous studies showed that CD38 deficiency protected mice from AngII-induced cardiac hypertrophy, ischemia/reperfusion injury, and cardiomyocyte senescence [20,21,22]. In the aspect of metabolism, CD38 deficiency suppressed adipogenesis and lipogenesis in adipose tissues and alleviated high-fat diet (HFD)-induced nonalcoholic fatty liver disease via the NAD^+^/Situins signaling pathway [23,24]. However, the roles of CD38 in diabetes-induced cardiomyopathy were not evaluated.

In the present study, we observed that the expression of CD38 was increased in heart tissues of type 2 diabetic mice. CD38 deficiency alleviated HFD plus streptozotocin (STZ)-induced diabetic cardiomyopathy. Knockdown of CD38 significantly reduced high glucose plus palmitic acid-induced cardiomyocytes dysfunction in H9C2 cells. Furthermore, we demonstrated that the protective roles of CD38 deficiency on diabetic cardiomyopathy were related to the inhibition of apoptosis and pyroptosis through activating Sirt3/FOXO3a signaling pathways.

## 2. Results

### 2.1. CD38 Expression Is Increased in Cardiac Tissues of Diabetic Mice

To examine the roles of CD38 in the hearts of type 2 diabetes, a mouse model of type 2 diabetes was prepared by feeding HFD and administering a single streptozotocin (STZ), as previously described [25]. The results showed that CD38 expression was significantly increased in protein and mRNA levels at 8 weeks following the onset of diabetes (Figure 1A,B,E). At the same time, the NAD^+^-dependent deacetylase Sirt3 protein and mRNA levels were down-regulated in hearts (Figure 1C–E). It was reported that cell death played important roles in diabetes-induced heart damage. In the present study, we detected the expression of the genes involved in apoptosis and pyroptosis. The results showed that the expression of pro-apoptotic gene Bax was increased in heart tissues of type 2 diabetic mice, and the ratio of Bax/Bcl2 was also increased (Figure 1F). In addition, we found that the NLRP3 and caspase-1 expressions were also up-regulated in heart tissues of diabetic mice compared with the control (Figure 1G). These results suggest that CD38 up-regulation is associated with diabetic cardiomyopathy and that CD38 may play an important role in this process.

### 2.2. CD38 Deficiency Reduces Bodyweight and Improves Glucose Metabolism In Vivo

CD38-KO mice were used to investigate a direct relationship between CD38 deficiency and diabetic cardiomyopathy. The efficiency of CD38 knockout was confirmed by Western blotting and QPCR. The results showed that CD38 was almost completely deleted in a variety of tissues (Figure 2A,B). In addition, the bodyweight of CD38 deficient mice with diabetes was decreased compared with the control group (Figure 2C), and CD38 deficiency significantly decreased the diabetes-induced increase in fasting glucose compared with control mice (Figure 2D). Moreover, we observed that the random blood glucose was reduced in the CD38-deficiency group compared with control mice under diabetes at 4, 6, and 8 weeks after HFD plus STZ injection (Figure 2E). Furthermore, the insulin resistance was also significantly improved in CD38-deficient mice fed with HFD plus STZ compared with CD38flox mice (Figure 2F,G). However, the glucose intolerance had no significant difference between the two groups (Figure 2H,I). These results indicate that CD38 deficiency improved glucose metabolism under diabetes.

### 2.3. CD38 Deficiency Improves Diabetes-Induced Cardiac Dysfunction

Next, we further determined whether the deficiency of CD38 protects the heart against diabetic cardiomyopathy. Cardiac function was measured via M-mode echocardiograms at 8 weeks after injection of STZ. As shown in Figure 3A, CD38 deficiency augmented diabetes-induced impairments of left ventricular (LV) ejection fraction and LV fractional shortening compared with control mice (Figure 3B,C), and LV mass was slightly decreased in CD38 knockout mice with no significant difference (Figure 3D). In addition, CD38 deficiency reduced the diabetes-induced increase in LDH activity in the serum compared with the control (Figure 3E). Moreover, CD38 deficiency remarkably reduced HFD plus STZ-induced increases of total cholesterol compared with the control group (Figure 3F). Taken together, these data indicate that CD38 deficiency prevented HFD + STZ-induced LV cardiac dysfunction.

### 2.4. CD38 Deficiency Ameliorates Diabetes-Induced Cardiac Pyroptosis In Vivo

An extensive selection of the literature indicates that cardiomyocyte death participates in the process of diabetic cardiomyopathy, including pyroptosis and apoptosis. In order to elucidate the mechanisms of CD38 deficiency protecting against diabetic cardiomyopathy, we first examined the expression of the genes involved in pyroptosis in heart tissues from CD38-deficient mice under normal and diabetic conditions. The results showed that the expression levels of NLRP3, caspase 1, IL-1β and IL-18 were increased in CD38flox mice with diabetes compared with the normal group, whereas CD38 deficiency significantly decreased the expression of these genes at the mRNA level (Figure 4A–D). Moreover, we also observed that the expression levels of NLRP3 and cleaved IL-1β were up-regulated in heart tissues of CD38flox mice with diabetes, but their expression levels were reduced in the CD38-deficient group compared with control mice under diabetes (Figure 4E–G). Furthermore, protein expression levels of the cleaved caspase 1 and GSDMD-N were significantly increased in diabetic mice, and were reversed in CD38 deficiency (Figure 4H–J). These results demonstrate that CD38 deficiency alleviated pyroptosis in diabetic hearts.

### 2.5. CD38 Deficiency Alleviates Diabetes-Induced Cardiac Apoptosis In Vivo

In addition to pyroptosis, in this study we also detected the expression levels of the genes involved in apoptosis. The results showed that the expressions of pro-apoptotic gene Bax at mRNA and protein levels were increased in heart tissues of diabetic mice, whereas CD38 deficiency significantly reduced the expression of Bax compared with the control group (Figure 5A,D). Conversely, the expression of anti-apoptotic gene Bcl2 was increased in CD38-deficient heart tissue, while there was no significant difference between the two groups under diabetes (Figure 5B). Meanwhile, CD38 deficiency also significantly decreased the diabetes-induced increase in the Bax/Bcl2 ratio, which was an indicator of apoptosis in mice (Figure 5C,E). Sirt3 is a NAD^+^-dependent protein deacetylase which is mainly located in mitochondria and deacetylates many targets such as FOXO3a. It was reported that Sirt3 deficiency exacerbated diabetic cardiac dysfunction [26]. In our study, we examined the expression levels of Sirt3 and its target gene FOXO3a in heart tissues. The results showed that the protein expression levels of Sirt3 and FOXO3a were reduced in diabetic mice, while CD38 deficiency significantly increased their expression (Figure 5F–I). Taken together, these results indicated that CD38 deficiency mediated the decrease in pyroptosis and apoptosis in diabetic cardiomyopathy, which might be associated with activating sirtuin-signaling pathways.

### 2.6. Knockdown of CD38 Protects High-Glucose- and High-Fat-Induced Cardiomyocyte Damage

To further clarify the protective roles of CD38 in diabetic cardiomyopathy in vitro, we examined the effects of CD38 on high glucose plus palmitic acid (HG/PA)-induced cardiomyocytic damage using CD38 knockdown H9C2 stable cell lines. The results showed that knockdown of CD38 significantly reduced the HG/PA-induced release of LDH in cardiomyocytes in vitro (Figure 6A). Decreased mitochondrial membrane potential is also an early signal of apoptosis and the mitochondrial membrane potential is usually decreased when cells are damaged. Our results showed that the mitochondrial membrane potential was markedly decreased after HG/PA treatment, exemplified by decreasing the ratio of red fluorescence of monomer/green fluorescence of aggregate, while CD38 knockdown remarkably increased it (Figure 6B). In addition, our results showed that the ROS production was increased by HG/PA, but there was no significant difference in CD38 knockdown H9C2 cells treated with HG/PA (Figure 6C). Moreover, we observed that the apoptosis was significantly increased after HG/PA treatment, while it decreased in the CD38 knockdown group under HG/PA (Figure 6D,E). All these results indicated that CD38 knockdown protected against cardiomyocyte damage induced by high glucose and high fat.

### 2.7. Knockdown of CD38 Suppresses Myocardial Pyroptosis and Apoptosis Induced by Hyperglycemia In Vitro

To further confirm whether CD38 affects HG/PA-induced pyroptosis and apoptosis in cardiomyocytes in vitro, we examined the expression levels of the genes involved in pyroptosis and apoptosis. The results showed that the mRNA expression levels of NLRP3, caspase 1, and IL-18 were decreased in CD38 knockdown cells after HG/PA treatment (Figure 7A–C). Moreover, the Western blot results also further confirmed that CD38 knockdown reduced the expressions of NLRP3 and cleaved IL-1β under high glucose plus palmitic acid (Figure 7D–F). In terms of apoptosis, we observed that the Bax expression was decreased, while the Bcl2 expression was increased at the mRNA level in CD38-knockdown H9C2 cells with HG/PA stimulation (Figure 8A,B). Meanwhile, the ratio of Bax/Bcl2 was significantly decreased in the CD38 knockdown group compared with the control group under HG/PA (Figure 8C). Similarly, Western blotting also showed that the Bcl2 expression was increased, but the ratio of Bax/Bcl2 was significantly decreased in CD38 knockdown H9C2 cells with HG/PA stimulation (Figure 8D,E). Furthermore, we found that the expression levels of Sirt3 and FOXO3a were significantly increased in CD38 knockdown H9C2 cells with or without HG/PA stimulation (Figure 8F–H). These results indicated that knockdown of CD38 decreased myocardial pyroptosis and apoptosis in vitro, in which the mechanism might be related to activating Sirt3/FOXO3a signaling pathways.

## 3. Discussion

Recently, *The Lancet* reported that type 2 diabetes accounted for nearly 90% of the approximately 537 million cases of diabetes in the worldwide [27]. Diabetes can lead to many complications, including macrovascular and microvascular diseases. Diabetic cardiomyopathy is one of the most important complications with high morbidity and mortality. CD38 is the main hydrolase of intracellular NAD^+^, and our previous studies showed that CD38 deficiency significantly protected against many cardiovascular diseases, such as cardiac hypertrophy, ischemia/reperfusion injury, and cardiomyocyte senescence [20,21,22]. In addition, our unpublished data showed that CD38 deficiency improved the cardiac functions and decreased the death rate of acute myocardial infarction in mice. These results suggest that CD38 plays important roles in cardiovascular diseases. However, it remains unknown whether CD38 participates in diabetes-induced cardiomyopathy and the underlying mechanisms. In the present study, we observed that CD38 was up-regulated in the heart tissue of mice under diabetes, and Sirt3, which is mainly located in mitochondria, was down-regulated. These results indicate that CD38/Sirt3 signaling pathways might play important roles in diabetic cardiomyopathy.

Our previous study reported that CD38 deficiency alleviated HFD-induced obesity [23]. In this study, the bodyweight of mice was lower in CD38 deficiency compared with the control group under HFD plus STZ. This result is consistent with our previous study. It was reported that CD38/cADPR-mediated Ca^2+^ signals played a key role in glucagon-induced gluconeogenesis in hepatocytes [28]. Our results showed that CD38 knockout mice had decreased blood glucose and significantly improved insulin resistance under type 2 diabetes. However, there was no significant difference in the glucose intolerance between the two groups. Our unpublished data showed that in type 1 diabetes, the glucose intolerance was improved, whereas the insulin resistance was not changed in CD38-deficient mice. The reason for this result is related to the different pathogenesis of the two types of diabetes. Diabetic cardiomyopathy is characterized by adverse structural remodeling and abnormal cardiac function [29]. Our results showed that CD38 deficiency improved cardiac function, exemplified by increased ejection fraction and shortening fraction under diabetes.

Although the pathophysiology of diabetic cardiomyopathy is complex and multifactorial, elevated cardiomyocyte death is a key contributor [30]. Studies indicate that pyroptosis is involved in the pathogenesis of cardiomyocyte injury, especially in diabetic cardiomyopathy [31]. It has been reported that the NLRP3 inflammasome-mediated classical pyroptosis signaling pathway is a key driving force in diabetic cardiomyopathy [32,33]. Recently, Yang et al. demonstrated that metformin suppressed NLRP3 inflammasome through AMPK/mTOR/autophagy pathways in diabetic cardiomyopathy [34]. In the present study, we found that the expression levels of the genes involved in pyroptosis, including NLRP3, caspase 1, IL-1β, and IL-18, were increased in heart tissues of type 2 diabetes, whereas the expression levels of these genes were remarkably decreased in the hearts of CD38-deficient mice with diabetes. In the caspase-1-dependent pathway, DAMP and PAMP act by cleaving GSDMD, whilst GSDMD-N directly mediates the pore formation of cell membranes. Our results showed that the active form of GSDMD-N expression was significantly reduced in CD38-deficient heart tissues and cardiomyocytes in vivo and in vitro, respectively.

In addition to pyroptosis, numerous studies have shown the apoptosis of cardiomyocytes to be a major event in the development of diabetic cardiomyopathy [35]. It was reported that there was a significant increase in cardiomyocyte apoptosis at 3 and 14 days after induction of type 1 diabetes in rats [36]. Under hyperglycemia, cardiomyocytes are also prone to apoptosis. Our previous study showed that apoptosis was decreased in CD38-deficient airway smooth muscle cells during hypoxia [37]. In this study, we found that the genes involved in apoptosis were inhibited in CD38-deficient hearts. The results from flow cytometry also showed the apoptosis was reduced in CD38 knockdown cardiomyocytes. Taken together, our results demonstrate that CD38 deficiency protected cardiomyocytes from HFD plus STZ or high glucose plus PA-induced damage through modulating pyroptosis and apoptosis in vivo and in vitro.

The sirtuins are a family of highly conserved NAD^+^-dependent deacetylases, including Sirt1-7. Among these, Sirt1 and Sirt3 have been most widely studied. Lots of studies have shown that Sirt3 plays important roles in preserving oxidative metabolism and increasing energy production (ATP synthesis) in mitochondria for maintaining proper function in the heart [38]. Recent studies also showed that Sirt3 had cardio-protective effects via anti-inflammatory, anti-fibrotic, and anti-apoptotic actions [39,40]. It has been reported that Sirt3 deficiency increased the expression of the inflammasome-related protein NLRP3, which promoted the recruitment of pro-inflammatory cells, caspase 1 activation, and pro-inflammatory cytokine secretion, ultimately exacerbating diabetic cardiomyopathy in these mice [41], and exacerbated diabetic cardiac dysfunction, mainly through Foxo3A-Parkin-mediated mitophagy [26]. FOXO3a is an important target protein of Sirt3. Our previous study showed that CD38 deficiency protected the heart from high-fat-diet-induced oxidative stress by activating Sirt3/FOXO3 pathways [18]. Numerous studies have indicated that there is a close relationship between pyroptosis and apoptosis [42,43]. These results demonstrate that Sirt3/FOXO3a signaling plays an important role in diabetic cardiomyopathy. In our study, we found that the expression levels of Sirt3 and FOXO3a were significantly increased in the hearts of CD38-deficient mice or cardiomyocytes under the condition of diabetes.

In conclusion, our study demonstrates that CD38 deficiency improved type 2 diabetes-induced cardiomyopathy via inhibiting pyroptosis and apoptosis in vivo and in vitro, in which the underlying mechanisms were mainly related to activating Sirt3/FOXO3a signaling pathways. Our findings should provide new insights into the elucidation of the mechanisms of cardiomyopathy induced by diabetes. And the new understanding of the role of CD38 in diabetic cardiomyopathy may have potential guiding significance for clinical treatment of the disease.

## 4. Materials and Methods

### 4.1. Mouse Procedure

CD38 flox mice (CD38flox, the second exon of CD38 gene was floxed by two loxP sites) were prepared by Cyagen (Suzhou, China). Male mice with global knockout of the CD38 gene (CD38KO) were obtained by crossing CD38floxed mice with EIIa-Cre mice (Cyagen, Suzhou, China) on a C57BL/6J background. Male CD38KO mice at the age of 6–8 weeks and age-matched CD38flox mice were used in this study. Experimental diabetic mice were induced by feeding a high-fat diet (60% HFD, D12492; Research Diets Inc., New Brunswick, NJ, USA) for 18 weeks followed by a single intraperitoneal injection of streptozotocin (STZ; pH 4.5; Sigma-Aldrich, Darmstadt, Germany) at 100 mg/kg in 0.1 mol/L of citrate acid buffer. Five days after STZ injection, mice with hyperglycemia (3 h fasting blood glucose levels ≥ 250 mg/dL) were referred as diabetic. Age-matched mice were fed with a normal diet for 18 weeks, followed by an injection of vehicle solution (0.1 mol/L of citrate acid buffer, pH 4.5) and were used as controls. The diabetic and control mice were then further fed with HFD or ND for 8 weeks, respectively. Bodyweight was measured every week after STZ induction. Random blood glucose was examined at 4, 6, and 8 weeks after HFD plus STZ injection. All animal experiments were approved by the Committee of Experimental Animals of Nanchang University.

### 4.2. Cell Culture and Treatment

H9C2 cells were cultured in Dulbecco’s modification of Eagle’s medium with 10% fetal bovine serum and 1% penicillin/streptomycin at 37 °C in an atmosphere of 5% CO_2_. The H9C2 stable cell line with CD38 knockdown was prepared as in a previous study [21]. At 70–80% confluence, the cells were cultured with the fresh media containing 5.5 mM D-glucose (control) or 33.3 mM D-glucose plus 0.5 mM PA (HG/PA) for 24–36 h.

### 4.3. Echocardiography

Mice were anesthetized with isoflurane. A Visual Sonics Vevo3100 Imaging System was used to measure heart rate and left ventricular dimensions from the 2D short axis under M-mode tracings. Cardiac functional parameters, including ejection fraction (EF), fractional shortening (FS), left ventricular internal diameter at end diastole (LVIDd) and left ventricular internal diameter at end systole (LVIDs) were calculated using the Vevo 3100 software.

### 4.4. Glucose Tolerance Test (GTT) and Insulin Tolerance Test (ITT)

For the GTT, mice were fasted for 16 h and then injected i.p. with D-glucose (1.5 g/kg). Glucose concentrations were determined using a glucometer (OneTouch Ultra; LifeScan, Inc. Malvern, PA, USA) in blood collected from the tail vein at the indicated time points. For the insulin tolerance test (ITT), 6 h fasted mice were injected i.p. with 0.75 U/kg insulin and tail vein blood glucose was then measured at the indicated times.

### 4.5. Biochemical Analysis

Lactate dehydrogenase (LDH) activity and the content of total cholesterol in serum or medium were measured using commercial kits following the manufacturers’ instructions. 

### 4.6. Mitochondrial Membrane Potential Assay

Mitochondrial membrane potential was detected using a JC-1 assay kit following the manufacturer’s protocol. Briefly, the cells were treated with 5.5 mM D-glucose or 33.3 mM D-glucose plus 0.5 mM PA for 36 h, then washed with PBS. The cells were stained with JC-1 dye for 20 min at 37 °C, and then washed with staining buffer twice. When detecting JC-1 monomer, the excitation light and emission light were set as 490 nm and 530 nm, respectively. When detecting JC-1 polymer, the excitation light was set to 525 nm, and the emission light was set to 590 nm.

### 4.7. Apoptosis Assay

For analysis of H9C2 cardiomyocytes after treatment with HG/PA, apoptosis assay was performed using the Annexin V-FITC Apoptosis Detection Kit (Dojindo, Shanghai, China). Cardiomyocytes were double stained with Annexin V-FITC and propidium iodide (PI) according to the manufacturer’s protocol. The cell apoptosis was detected by flow cytometry (CytoFLEX, Beckman Coulter, Brea, CA, USA).

### 4.8. ROS Detection

The content of ROS production in cells was detected using H2DCF-DA (Sigma-Aldrich, Darmstadt, Germany) as previously described [22]. Briefly, the cells were treated with 5.5 mM D-glucose or 33.3 mM D-glucose plus 0.5 mM PA for 36 h, then washed with PBS. The cells were digested with trypsin and incubated with 10 mM H2DCF-DA for 30 min at 37 °C with an atmosphere of 5% CO_2_, keeping out light. Then, the cells were washed with PBS and the fluorescence was detected with an automatic microplate reader at wavelengths of 488 nm (excitation) and 520 nm (emission).

### 4.9. Total RNA Extraction and Real-Time PCR

Total RNA from mouse tissue and cultured cells were isolated using Trizol reagent (Invitrogen, Waltham, MA, USA) following the manufacturer’s protocol. The RNA concentration was measured via Nano 2000 (Thermo Fisher, Waltham, MA, USA). Then, RNA was reverse transcribed using the Takara high-capacity cDNA synthesis kit and used in quantitative PCR reactions containing SYBR green fluorescent dye (Takara, Dalian, China). Quantitative PCR was performed using the ABI-ViiA7 PCR machine. Relative expression of mRNA was determined using the delta CT method with GAPDH as the housekeeping gene. Q-PCR primer sequences are listed in Table 1.

### 4.10. Western Blotting

Cardiac tissue samples and H9C2 cells were homogenized in lysis buffer. Lysates were centrifuged at 12,000× *g* for 10 min at 4 °C. Then, the protein concentration was quantified using a commercial assay with BSA as standard. Lysates were separated by SDS-PAGE, and electro-transferred to a PVDF membrane (Millipore, Burlington, MA, USA). The membranes were blocked for 1.5 h at room temperature in Tris-buffered saline (pH 7.6) containing 5% non-fat dry milk, and then immunoblotted overnight at 4 °C with primary antibody. Secondary antibodies were applied for 1 h at room temperature. The expression of protein was visualized using enhanced chemi-luminescence Western blotting detection

Reagents. The following antibodies were used for Western blot analysis: CD38 (R&D), Sirt3 (CST), Sirt1 (Millipore), NLRP3 (CST), cleaved caspase-1 (CST), cleaved IL-1β (CST), GSDMD (CST), Bax (CST), Bcl2 (CST) and FOXO3a (abcam).

### 4.11. Statistical Analysis

The in vitro experiments were repeated at least three times. Data were presented as mean ± SEM. Student’s *t* test and one-way analysis of variance (ANOVA) were used for comparation of two or multiple groups, respectively, with SPSS17.0. Statistical significances were shown as * *p* < 0.05, ** *p* < 0.01, and *** *p* < 0.001.

## Figures and Tables

**Figure 1 ijms-24-16008-f001:**
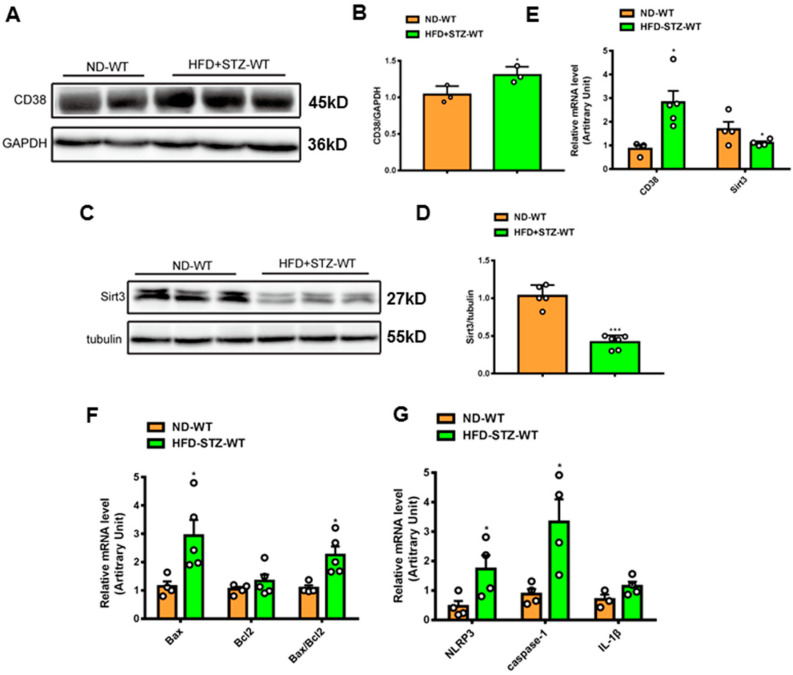
CD38 expression was increased in hearts of mouse models of type 2 diabetes. (**A**,**B**) Western blotting and associated quantitative analysis of heart CD38 protein expression at 8 weeks after HFD plus STZ injection. (**C**,**D**) Western blotting and associated quantitative analysis of heart Sirt3 protein expression at 8 weeks after HFD plus STZ injection. (**E**) QPCR analysis of heart CD38 and Sirt3 mRNA expression at 8 weeks after HFD plus STZ injection. (**F**,**G**) QPCR analysis of genes involved apoptosis and pyroptosis mRNA expression at 8 weeks after HFD plus STZ injection. Data are shown as means ± SEM, * *p* < 0.05 and *** *p* < 0.001, *n* = 3~6 per group. The circles represented the number.

**Figure 2 ijms-24-16008-f002:**
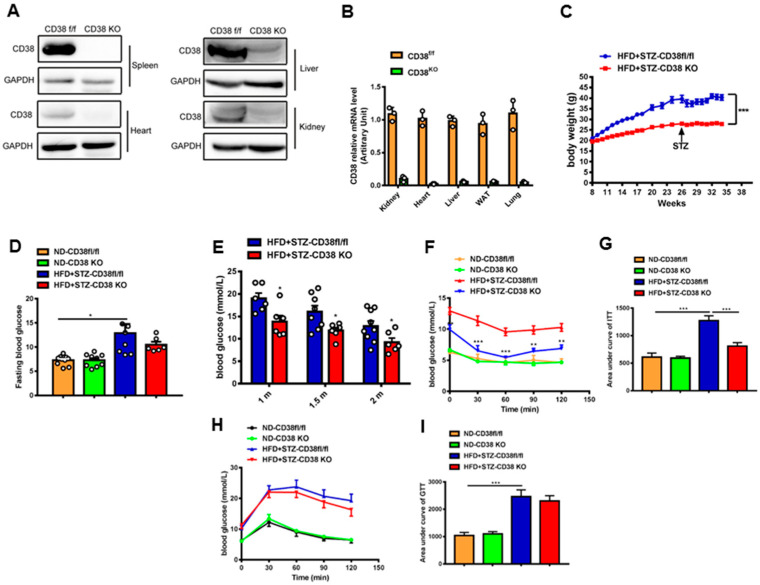
CD38 deficiency reduced bodyweight and improved glucose metabolism. (**A**,**B**) The efficiency of CD38 knockout was confirmed by Western blotting and QPCR. (**C**) The bodyweights of the CD38flox and CD38 knockout mice which were fed with HFD plus STZ injection were measured every week. (**D**) The fasting glucose was measured at 8 weeks after HFD plus STZ injection. (**E**) The random glucose was measured at 4, 6, and 8 weeks after HFD plus STZ injection. (**F**,**G**) Insulin tolerance test (ITT) was carried out in CD38flox and CD38 knockout mice at 6 weeks after ND or HFD plus STZ injection after 6 h fasting. (**H**,**I**) Glucose tolerance test (GTT) was carried out in CD38flox and CD38 knockout mice at 7 weeks after ND or HFD plus STZ injection after 16 h fasting. Data are shown as means ± SEM, * *p* < 0.05, ** *p* < 0.01 and *** *p* < 0.001, *n* = 6~9 per group.

**Figure 3 ijms-24-16008-f003:**
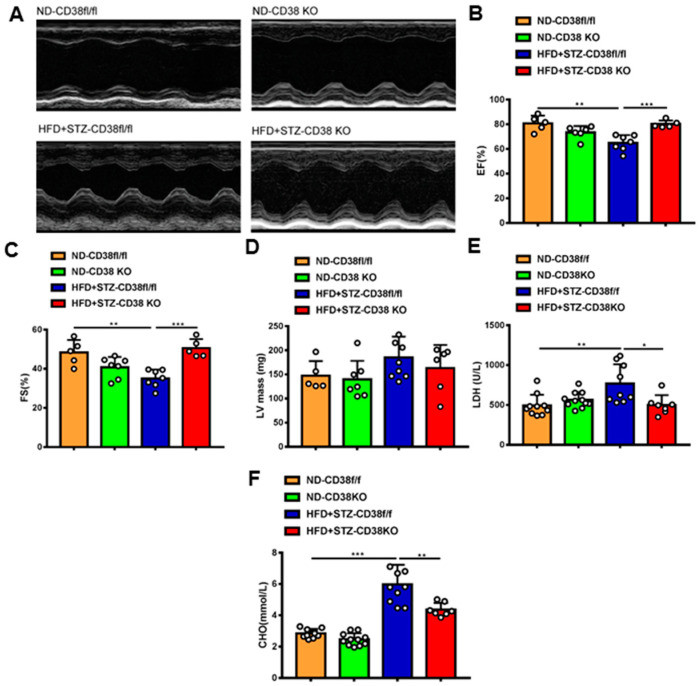
CD38 deficiency alleviated diabetes-induced cardiac dysfunction. (**A**) Representative echocardiograms obtained from CD38flox and CD38KO mice subjected to ND or HFD plus STZ injection at 8 weeks. (**B**–**D**) Echocardiographic measurements of left ventricular ejection fraction (EF), fractional shortening (FS), and left ventricular mass were measured in CD38flox and CD38KO mice subjected to ND or HFD plus STZ injection at 8 weeks. (**E**,**F**) The LDH activity and total cholesterol in serum were examined in CD38flox and CD38KO mice subjected to ND or HFD plus STZ injection at 8 weeks. Data are shown as means ± SEM, * *p* < 0.05, ** *p* < 0.01 and *** *p* < 0.001, *n* = 5~11 per group.

**Figure 4 ijms-24-16008-f004:**
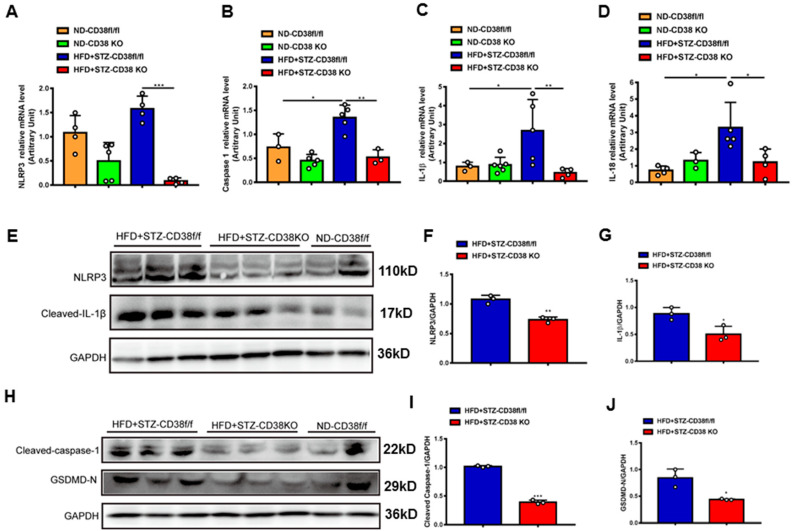
CD38 deficiency ameliorated diabetes-induced cardiac pyroptosis in vivo. (**A**–**D**) QPCR analysis of NLRP3, caspase1, IL-1β, and IL-18 mRNA levels in cardiac tissues of CD38flox and CD38KO mice subjected to ND or HFD plus STZ injection at 8 weeks. (**E**–**G**) Western blotting and associated quantitative analysis of heart NLRP3 and cleaved IL-1β protein expression in cardiac tissues of CD38flox and CD38KO mice subjected to ND or HFD plus STZ injection at 8 weeks. (**H**–**J**) Western blotting and associated quantitative analysis of heart cleaved caspase-1 and GSDMD-N protein expression in cardiac tissues of CD38flox and CD38KO mice subjected to ND or HFD plus STZ injection at 8 weeks. Data are shown as means ± SEM, * *p* < 0.05, ** *p* < 0.01 and *** *p* < 0.001, *n* = 3~5 per group.

**Figure 5 ijms-24-16008-f005:**
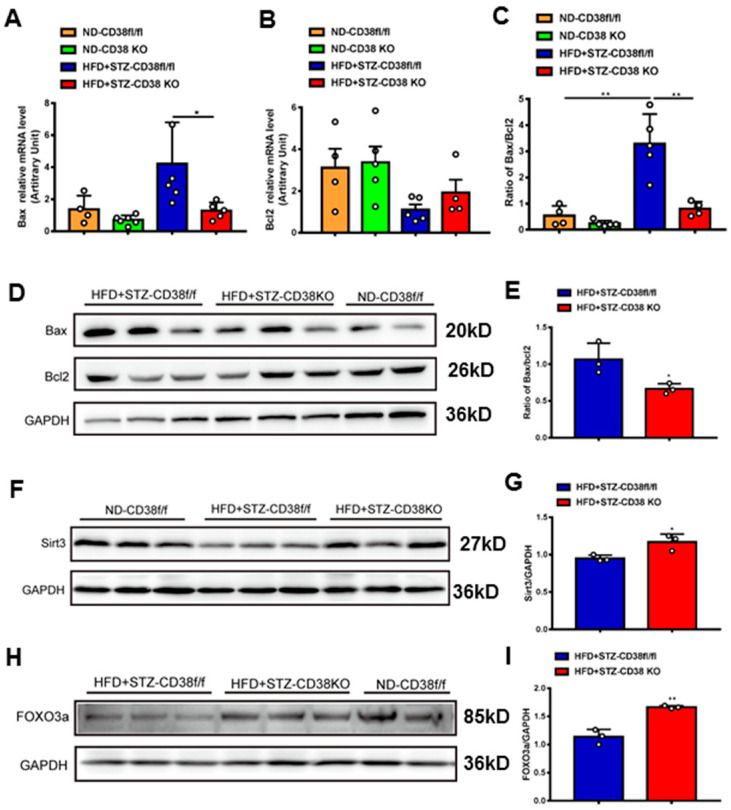
CD38 deficiency decreased diabetes-induced cardiac apoptosis. (**A**–**C**) The expression levels of apoptosis genes such as Bax, Bcl2 and the ratio of Bax/Bcl2 were quantitatively analyzed via QPCR in heart tissue from CD38flox and CD38KO mic subjected to ND or HFD plus STZ injection at 8 weeks. (**D**,**E**) The protein expression levels of Bax, Bcl2 in heart tissue from CD38flox and CD38KO mice were confirmed by Western blot analysis. (**F**,**G**) Western blotting and associated quantitative analysis of heart Sirt3 protein expression in cardiac tissues of CD38flox and CD38KO mice. (**H**,**I**) Western blotting and associated quantitative analysis of heart FOXO3a protein expression in cardiac tissues of CD38flox and CD38KO mice. Data are shown as means ± SEM, * *p* < 0.05 and ** *p* < 0.01, *n* = 3~6 per group.

**Figure 6 ijms-24-16008-f006:**
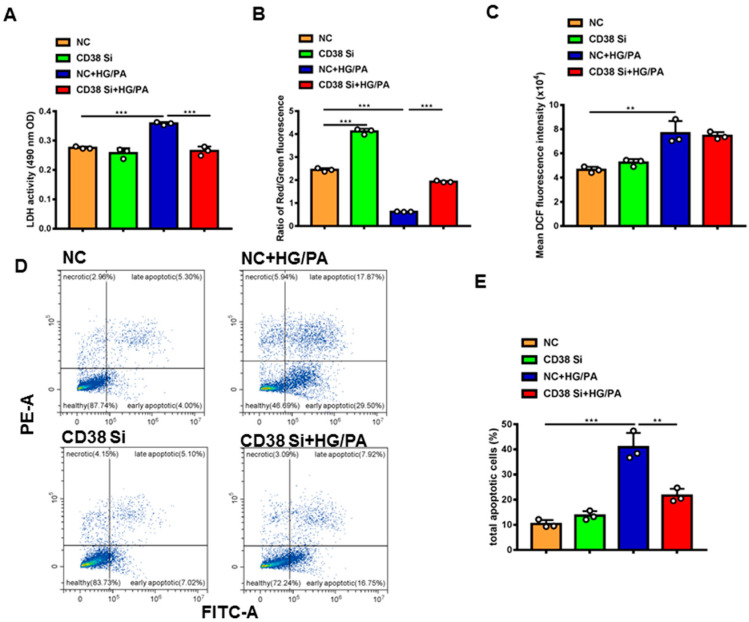
Knockdown of CD38 protected against high-glucose- and high-fat-induced cardiomyocyte damage. (**A**) The LDH activity in supernatant of medium of control and CD38 knockdown H9C2 cells after high-glucose and high-fat treatment. (**B**) Mitochondrial membrane potential was detected with JC-1 in control and CD38 knockdown H9C2 cells. (**C**) The ROS production was detected with an H2DCF-DA probe in control and CD38 knockdown H9C2 cells. (**D**) Representative diagram of cell apoptosis of H9c2 cells detected by flow cytometry using Annexin V-FITC/PI double staining. (**E**) The apoptotic rate shown in the bar graph was calculated. Data are shown as means ± SEM, ** *p* < 0.01 and *** *p* < 0.001, *n* = 3 per group.

**Figure 7 ijms-24-16008-f007:**
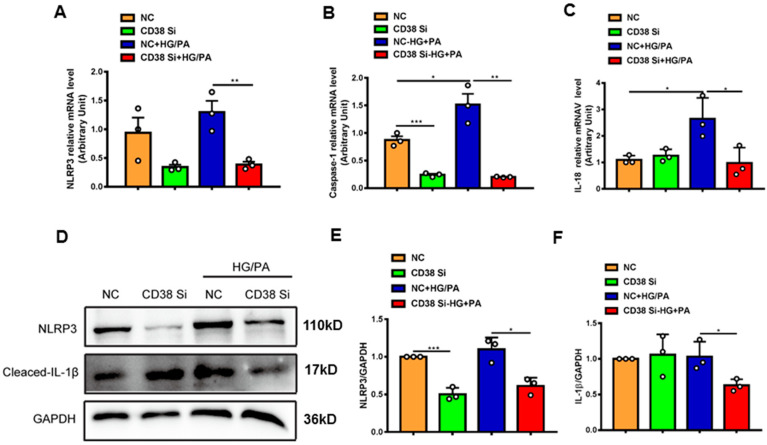
Knockdown of CD38 suppresses myocardial pyroptosis induced by hyperglycemia in vitro. (**A**–**C**) QPCR analysis of NLRP3, caspase1, and IL-18 mRNA levels in control and CD38 knockdown H9C2 cells challenged with or without HG and PA. (**D**–**F**) Western blotting and associated quantitative analysis of NLRP3 and cleaved IL-1β protein expression in control and CD38 knockdown H9C2 cells challenged with or without HG and PA. Data are shown as means ± SEM, * *p* < 0.05 and ** *p* < 0.01 and *** *p* < 0.001, *n* = 3 per group.

**Figure 8 ijms-24-16008-f008:**
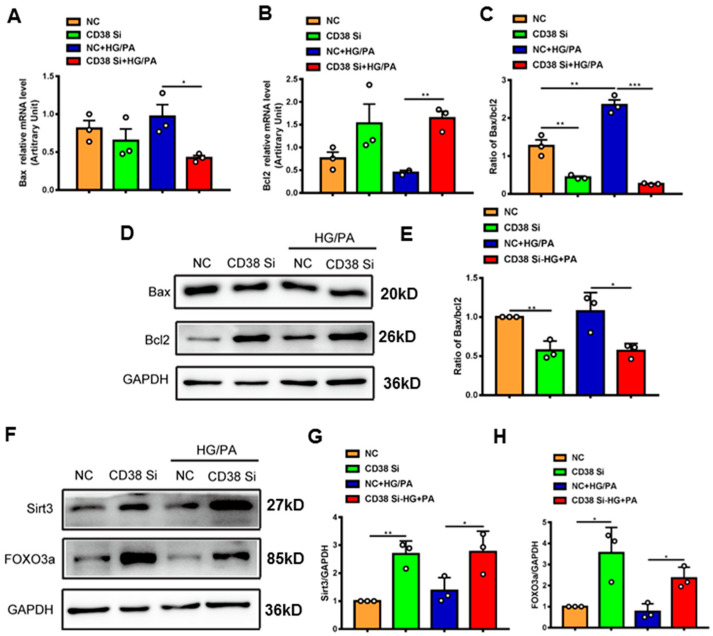
Knockdown of CD38 suppresses myocardial apoptosis induced by hyperglycemia in vitro. (**A**–**C**) The expression levels of apoptosis genes such as Bax, Bcl2 and the ratio of Bax/Bcl2 were quantitatively analyzed by QPCR in control and CD38 knockdown H9C2 cells challenged with or without HG and PA. (**D**,**E**) Western blotting and associated quantitative analysis of Bax and Bcl2 protein expression in control and CD38 knockdown H9C2 cells challenged with or without HG and PA. (**F**–**H**) Western blotting and associated quantitative analysis of Sirt3 and FOXO3a protein expression in control and CD38 knockdown H9C2 cells challenged with or without HG and PA. Data are shown as means ± SEM, * *p* < 0.05, ** *p* < 0.01 and *** *p* < 0.001, *n* = 3 per group.

**Table 1 ijms-24-16008-t001:** QPCR primers used in this study.

Gene Name	Forward	Reverse
m CD38	CTGCCAGGATAACTACCGACCT	CTTTCCCGACAGTGTTGCTTCT
m Sirt3	ATCCCGGACTTCAGATCCCC	CAACATGAAAAAGGGCTTGGG
m Bax	AGGATGCGTCCACCAAGAAG	CCATATTGCTGTCCAGTTCATCTC
m Bcl2	ATGTGTGTGGAGAGCGTCAA	AGAGACAGCCAGGAGAAATCA
m NLRP3	GTGGAGATCCTAGGTTTCTCTG	CAGGATCTCATTCTCTTGGATC
m Caspase1	ACACGTCTTGCCCTCATTATCT	ATAACCTTGGGCTTGTCTTTCA
m IL-1β	CCCTGCAGCTGGAGAGTGTGG	TGTGCTCTGCTTGAGAGGTGCT
m IL-18	ACAACCGCAGTAATACGGAGCA	TGTGCTCTGCTTGAGAGGTGCT
m TGFβ	ACTGGAGTTGTACGGCAGTG	GGGGCTGATCCCGTTGATT
m αSMA	ACTGGGACGACATGGAAAAG	GTTCAGTGGTGCCTCTGTCA
r NLRP3	GTGGAGATCCTAGGTTTCTCTG	CAGGATCTCATTCTCTTGGATC
r Caspase1	GAGCTGATGTTGACCTCAGAG	CTGTCAGAAGTCTTGTGCTCTG
r IL-18	ACAACCGCAGTAATACGGAGCA	TGTGCTCTGCTTGAGAGGTGCT
r Bax	GGGTGGCAGCTGACATGTTT	GCCTTGAGCACCAGTTTGC
r Bcl2	GTTGCAGTCACCGGATTCCT	CGGA GGTGGTGTGAATCCA
GAPDH	AGCCAAAAGGGTCATCATCT	GGGGCCATCCACAGTCTTCT

## Data Availability

The data are available on request from the authors.

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
