# Peer review of "CD38 Deficiency Alleviates Diabetic Cardiomyopathy by Coordinately Inhibiting Pyroptosis and Apoptosis"

_ijms, 2023, doi:10.3390/ijms242116008_

Round 1
Reviewer 1 Report
Comments and Suggestions for Authors
The study titled "CD38 deficiency alleviates diabetic cardiomyopathy by coordinately inhibiting pyroptosis and apoptosis" by Ling-Fang Wang et al., is interesting. The effect of CD38 is well known as a therapeutic target for cardiovascular diseases and this study addresses whether CD38 knockdown can protect from the development of Diabetic cardiomyopathy under high fat and streptozotocin-induced diabetes. The study design is appropriate, and the results seem convincing. However, the following are few concerns that I have with the study.
1) Line 89 in results section 2.1. The statement "These results suggested that CD38 played an important role in the role in the development of diabetic cardiomyopathy" is not valid at the place. For the results it is not conclusive. It is only showing CD38 was upregulated in the HFD+STZ WT mice in RNA and protein level.
2) None of the western blot figures have molecular weight markers.
3)The statement in result section 2.3 line-132 also is not correct. The CD38 deficiency is from birth itself and therefore it is better to say that CD38 deficiency prevented HFD+STZ-induced LV cardiac dysfunction rather than restored LV cardiac function in type 2 diabetic mice.
4) Figure 4, E and H, the blots are not good with GAPDH bands looking underdeveloped due to substrate depletion. Please redo this or replace the representative blots.
5) it would be better to include the ND CD38KO group also in the Figure 4 and Figure 5 western blots to see what the baseline difference of Cd38 deficiency is.
Author Response
Comments and Suggestions for Authors
The study titled "CD38 deficiency alleviates diabetic cardiomyopathy by coordinately inhibiting pyroptosis and apoptosis" by Ling-Fang Wang et al., is interesting. The effect of CD38 is well known as a therapeutic target for cardiovascular diseases and this study addresses whether CD38 knockdown can protect from the development of Diabetic cardiomyopathy under high fat and streptozotocin-induced diabetes. The study design is appropriate, and the results seem convincing. However, the following are few concerns that I have with the study.
1) Line 89 in results section 2.1. The statement "These results suggested that CD38 played an important role in the role in the development of diabetic cardiomyopathy" is not valid at the place. For the results it is not conclusive. It is only showing CD38 was upregulated in the HFD+STZ WT mice in RNA and protein level.
Response: Thank you very much for your careful review. We have revised our manuscript in the revised version in section 2.1. The statement "These results suggested that CD38 played an important role in the role in the development of diabetic cardiomyopathy" has been replaced by "These results suggested that CD38 upregulation is associated with diabetic cardiomyopathy and that CD38 may play an important role in this process".
2) None of the western blot figures have molecular weight markers.
Response: Thank you very much for your good suggestions. We have added molecular weight markers in all western blot figures in the revised version.
3)The statement in result section 2.3 line-132 also is not correct. The CD38 deficiency is from birth itself and therefore it is better to say that CD38 deficiency prevented HFD+STZ-induced LV cardiac dysfunction rather than restored LV cardiac function in type 2 diabetic mice.
Response: Thank you very much for your good suggestions. In the revised manuscript the statement "these data indicated that CD38 deficiency restored LV cardiac function in type 2 diabetic mice" has been replaced by "these data indicated that CD38 deficiency prevented HFD+STZ-induced LV cardiac dysfunction".
4) Figure 4, E and H, the blots are not good with GAPDH bands looking underdeveloped due to substrate depletion. Please redo this or replace the representative blots.
Response: Thank you very much for your good suggestions. We have redone the experiment and replaced the representative blots in Figure 4E and H.
5) it would be better to include the ND CD38KO group also in the Figure 4 and Figure 5 western blots to see what the baseline difference of Cd38 deficiency is.
Response: Thank you very much for your good suggestions. We redid the western blotting experiments in ND group and found the indicators in Figure 4 and Figure 5 did not change much under ND conditions. The results were uploaded in attachment.

Reviewer 2 Report
Comments and Suggestions for Authors
Interesting work - the authors undertook the search for the relationship between CD38 and diabetic cardiomyopathy. The work was properly designed and carried out. Results clearly presented and well discussed.
In the manuscript, I propose to present the background more - please consider adding a few paragraphs in the introduction regarding the interaction through the pathway of the youngest mediators, nitric oxide and carbon monoxide. please also emphasize in a few sentences the role (importance of modulation) of the guanylate cyclase pathway in diseases such as heart failure or pulmonary hypertension. Adding this aspect will allow for a seamless connection between the facts regarding these common diseases in clinical practice and the experiment described in the manuscript.
One final note. Personally, because I work with such a group of patients on a daily basis, DCM is an abbreviation commonly used for dilated cardiomyopathy (DCM). I suggest either not abbreviating diabetic CM in the text (this is a minor note that does not need to be implemented, as there are works using this abbreviation.)
I also suggest adding one more sentence in the conclusion. Each study is intended to increase knowledge about a given topic. This manuscript undoubtedly does too. Please add information whether the authors believe that the new understanding of the role of CD38 may change potential therapeutic treatment? diagnostic? These works will also be read by doctors, both practitioners of clinical and experimental medicine :-)
Author Response
Comments and Suggestions for Authors
Interesting work - the authors undertook the search for the relationship between CD38 and diabetic cardiomyopathy. The work was properly designed and carried out. Results clearly presented and well discussed.
In the manuscript, I propose to present the background more - please consider adding a few paragraphs in the introduction regarding the interaction through the pathway of the youngest mediators, nitric oxide and carbon monoxide. please also emphasize in a few sentences the role (importance of modulation) of the guanylate cyclase pathway in diseases such as heart failure or pulmonary hypertension. Adding this aspect will allow for a seamless connection between the facts regarding these common diseases in clinical practice and the experiment described in the manuscript.
Response: Thank you very much for your suggestions. In our revised manuscript, we have added the nitric oxide and carbon monoxide mediators in our introduction, especially their roles in diabetic cardiomyopathy. Moreover, we also highlighted the role of guanylate cyclase pathway in diabetic cardiomyopathy.
One final note. Personally, because I work with such a group of patients on a daily basis, DCM is an abbreviation commonly used for dilated cardiomyopathy (DCM). I suggest either not abbreviating diabetic CM in the text (this is a minor note that does not need to be implemented, as there are works using this abbreviation.)
Response: Thank you very much for your good suggestions. We have revised our manuscript and the abbreviation DCM or diabetic CM were all replaced by Diabetic cardiomyopathy.
I also suggest adding one more sentence in the conclusion. Each study is intended to increase knowledge about a given topic. This manuscript undoubtedly does too. Please add information whether the authors believe that the new understanding of the role of CD38 may change potential therapeutic treatment? diagnostic? These works will also be read by doctors, both practitioners of clinical and experimental medicine :-)
Response: Thank you very much for your good suggestions. We have revised our manuscript and highlighted the significance of our work for the treatment of diabetic cardiomyopathy.
Round 2
Reviewer 1 Report
Comments and Suggestions for Authors
The authors have satisfactorily revised the manuscript.
Reviewer 2 Report
Comments and Suggestions for Authors
the authors significantly modified the manuscript. in my opinion it may be considered for publication